# Robust Cross-modal Alignment Learning for Cross-Scene Spatial Reasoning and Grounding

**Yanglin Feng**[1], **Hongyuan Zhu**[2], **Dezhong Peng**[1,3], **Xi Peng**[1], **Xiaomin Song**[4], **Peng Hu**[1*]

[1]College of Computer Science, Sichuan University, Chengdu, China.
[2]Institute for Infocomm Research (I$^2$R), A*STAR, Singapore.
[3]Tianfu Jincheng Laboratory, Chengdu, China.
[4]Sichuan National Innovation New Vision UHD Video Technology Co., Ltd., Chengdu, China.
`fcyzfyl@163.com, hongyuanzhu.cn@gmail.com, pengdz@scu.edu.cn,`
`pengx.gm@gmail.com, songxiaomin@uptcsc.com, penghu.ml@gmail.com`

## Abstract

Grounding target objects in 3D environments via natural language is a fundamental capability for autonomous agents to successfully fulfill user requests. Almost all existing works typically assume that the target object lies within a known scene and focus solely on in-scene localization. In practice, however, agents often encounter unknown or previously visited environments and need to search across a large archive of scenes to ground the described object, thereby invalidating this assumption. To address this, we reveal a novel task called Cross-Scene Spatial Reasoning and Grounding (CSSRG), which aims to locate a described object anywhere across an entire collection of 3D scenes rather than predetermined scenes. Due to the difference from existing 3D visual grounding, CSSRG poses two challenges: the prohibitive cost of exhaustively traversing all scenes and more complex cross-modal spatial alignment. To address the challenges, we propose a **Cro**ss-Scene 3D Object **Re**asoning Framework (CoRe), which adopts a matching-then-grounding pipeline to reduce computational overhead. Specifically, CoRe consists of i) a Robust Text-Scene Aligning (RTSA) module that learns global scene representations for robust alignment between object descriptions and the corresponding 3D scenes, enabling efficient retrieval of candidate scenes; and ii) a Tailored Word-Object Associating (TWOA) module that establishes fine-grained alignment between words and target objects to filter out redundant context, supporting precise object-level reasoning and alignment. Additionally, to benchmark CSSRG, we construct a new CrossScene-RETR dataset and evaluation protocol tailored for cross-scene grounding. Extensive experiments across four multimodal datasets demonstrate that CoRe dramatically reduces computational overhead while showing superiority in both scene retrieval and object grounding. Code is available at `https://github.com/Yangl1nFeng/CoRe`.

## 1 Introduction

Grounding objects in 3D environments with natural language has emerged as a pivotal advancement in multimodal artificial intelligence, enhancing object understanding and interaction of autonomous agents. Building on this progress, recent developments in 3D Visual Grounding (3DVG) [1, 2, 3] and Group-wise 3D Object Grounding (GNL3D) [4] have further demonstrated the capability of agents to locate objects accurately within several given scenes using linguistic cues. However, in real-world scenarios, users may inquire about where specific events occurred or seek to distinguish

---

*Corresponding author.

39th Conference on Neural Information Processing Systems (NeurIPS 2025)

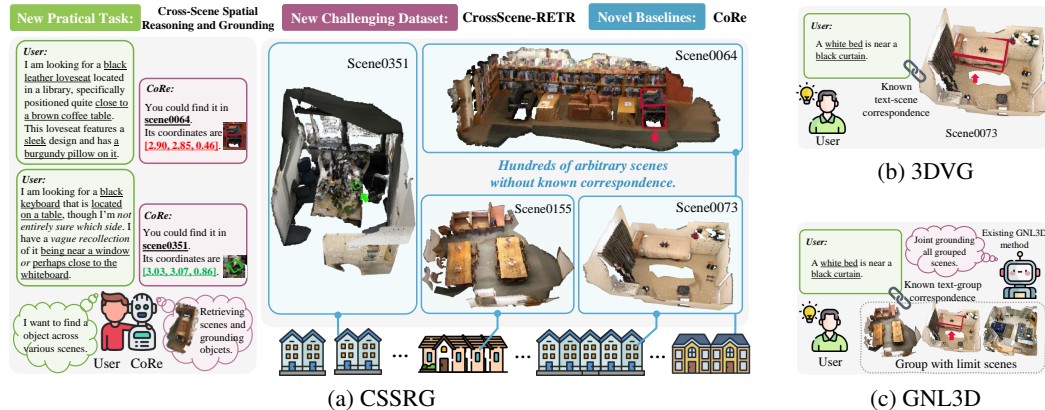

Figure 1: Overview of our proposed Cross-Scene Spatial Reasoning and Grounding (CSSRG) and comparisons with similar tasks. (a) illustrates our CSSRG. (b) and (c) show the illustrations of 3D Visual Grounding (3DVG) and Group-wise 3D Object Grounding (GNL3D), respectively.

objects across various places. Addressing such queries requires agents to reason over numerous memory scenes from their long deployment history. To concretize this requirement, we propose and study a more general task, Cross-Scene Spatial Reasoning and Grounding (CSSRG), which requires locating described objects anywhere across an entire collection of 3D scenes. The scene archive is not a collection of disjoint scenes, but rather represents the locations traversed by agents within a large-scale continuous map [5]. In contrast to 3DVG and GNL3D, which assume the availability of predefined corresponding scenes, CSSRG seeks to unlock the text-scene correspondence for general object localization, as shown in Figure 1. Benefiting from this, CSSRG would serve as a technical foundation for building-scale indoor navigation [6, 7] and task planning [8, 9], thereby enabling broader applications in smart homes and robotics.

However, directly applying existing methods to tackle CSSRG is infeasible and would face substantial challenges of high computational costs and low performance, as shown in Figure 2. Specifically, although existing 3DVG and GNL3D approaches can ground objects within known scenes, they either require meticulous scene-by-scene reasoning or concatenating all scenes for joint grounding, resulting in prohibitive computational and memory costs. To address this, an intuitive alternative is to use efficient cross-modal matching methods to retrieve the most relevant scenes and then apply 3DVG within

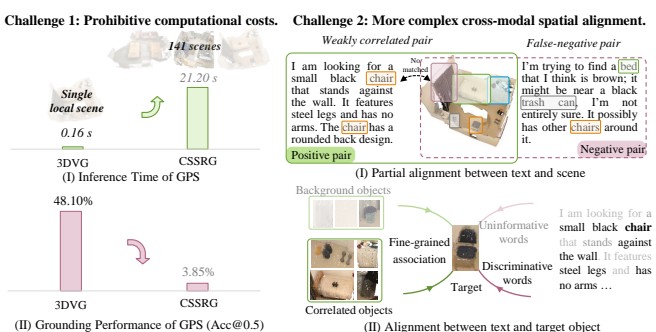

Figure 2: Task-specific challenges faced by CSSRG are illustrated with the VisTA method [10] on the ScanRefer dataset.

them, reducing the scene traversal time remarkably. It requires cross-modal spatial alignment of the query texts with both relevant scenes and target objects, which is by no means an easy task. To be specific, object descriptions typically focus on target objects instead of an entire scene, resulting in a partial alignment problem.

To overcome these challenges, we propose a **Cro**ss-Scene 3D Object **Re**asoning Framework (CoRe), which efficiently handles CSSRG via two key modules: a Robust Text-Scene Aligning module (RTSA) for scene matching and a Tailored Word-Object Associating module (TWOA) for object grounding. More specifically, our RTSA first aggregates multimodal fine-grained features into global representations, facilitating convenient text-scene alignment. However, the partial alignment between texts and scenes implies the presence of a certain number of mismatched pairs, leading to degradation of alignment performance. To address this issue, our RTSA adopts an innovative complementary learning paradigm that adaptively pushes apart negative pairs in the common space to robustly establish text-scene alignment, thereby achieving reliable scene matching. Moreover, our TWOA presents a novel Screening Attention mechanism (ScA) to construct the association between target objects and text words, enabling spatial reasoning from text to objects within the scenes. Specifically,

ScA progressively prunes low-attention word-object associations in a coarse-to-fine manner and dynamically integrates the contextually relevant retentions, seeking non-redundant alignment between textual descriptions and target objects.

Our CoRe outperforms existing methods in CSSRG, achieving superior scene-matching and object-grounding performance compared to general baselines, as demonstrated in Table 4. These results confirm that achieving complex cross-modal spatial alignment remains a significant challenge under the constraints of existing datasets. To further address this issue, we present a CrossScene-RETR benchmark, which includes discriminative object descriptions and a tailored evaluation protocol specifically designed for CSSRG. As demonstrated in Table 2, the use of comprehensive descriptions brings robust performance gains for CSSRG, supporting a more reliable evaluation and embracing practical applicability. In summary, our contributions are as follows:

- We extend the 3D visual grounding task to the more general Cross-Scene Spatial Reasoning and Grounding (CSSRG) task, which aims to ground a described object anywhere across an entire collection of 3D scenes instead of predetermined scenes.
- We propose the novel two-stage **Cro**ss-Scene 3D Object **Re**asoning Framework (CoRe), following a matching-then-grounding paradigm to effectively mitigate computational costs. CoRe includes a Robust Text-Scene Aligning module (RTSA) for robust scene matching and a Tailored Word-Object Associating module (TWOA) for object grounding.
- We present the CrossScene-RETR dataset to facilitate complex cross-modal spatial alignment in the data aspect, offering a comprehensive evaluation for CSSRG.
- Extensive experiments on four multimodal datasets demonstrate the superiority and effectiveness of our CoRe in CSSRG, remarkably outperforming state-of-the-art baselines.

## 2 Related Works

### 2.1 3D Visual Grounding

In recent years, the application potential of vision and language has been constantly explored, attracting significant attention [2, 10, 11]. As one of the primary tasks, 3D Visual Grounding (3DVG) has also gained considerable interest. More specifically, numerous methods [12, 13, 14, 15, 16] attempt to fully explore the specific information (*e.g.*, viewpoints, text graphs, *etc*.) from two modalities to boost the performance. Others [17, 18] attempt to introduce 2D pre-trained knowledge to achieve more comprehensive scene understanding by employing multi-view images. In addition, several approaches [10, 19, 20] proposed general frameworks to tackle multiple 3D vision and language tasks, aiming to break down the barriers between tasks and achieve complementary performance gains. Recently, a study [4] has attempted to expand the scope of grounding to point-cloud groups composed of a limited number of scenes. However, it remains limited to scene-by-scene inference, rendering object reasoning across a large-scale scene set challenging. To address this issue, this paper proposes a general approach to reason target objects from numerous scenes efficiently.

### 2.2 Cross-modal Matching

Cross-Modal Matching (CMM) [21, 22, 23] has received widespread attention in recent years, which aims to match the relevant results for given queries across different modalities. Due to its substantial role in multi-modal data management and pattern discovery, it is widely applied across various modalities and achieves notable success, *e.g.*, image-text matching [24, 25, 26, 27, 28], text-video matching [29], Infrared-visible matching [30], *etc*. Typically, most CMM methods focus on mapping heterogeneous data into common representations for modality-invariant matching. Specifically, CMM methods could be grouped into fine-grained and coarse-grained methods. 1) Fine-grained approaches [31, 32] aim to capture more nuanced cross-modal semantic associations by focusing on fine-grained features, such as image regions, text words, *etc*. 2) Coarse-grained methods [33, 34] aggregate fine-grained features into holistic representations, seeking straightforward alignment by employing the cross-modal contrast constraints [35, 36, 37]. However, existing methods aim at establishing correspondence between instances, making it difficult to precisely align fine-grained objects with whole texts in the Cross-Scene Spatial Reasoning and Grounding task.

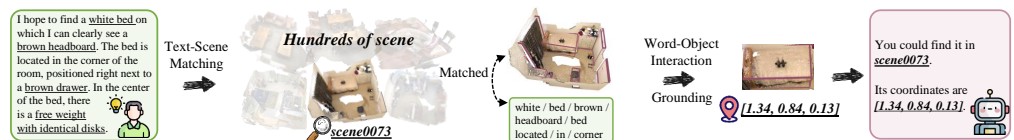

Figure 3: The solution pipeline for the Cross-Scene Spatial Reasoning and Grounding (CSSRG) task.

# 3 Task: Cross-Scene Spatial Reasoning and Grounding

In this paper, we present a Cross-Scene Spatial Reasoning and Grounding (CSSRG) task, with its pipeline illustrated in Figure 3. This task involves grounding a 3D object from a scene archive that relies only on a query description. More specifically, CSSRG requires efficiently retrieving the most relevant scene containing the target 3D object from hundreds of scene candidates based on a user description. Subsequently, all objects in the matched scene and all words in the description must undergo fine-grained fusion to ultimately reason about the target object. During the abovementioned process, CSSRG introduces task-specific challenges, as illustrated in Figure 2, which could be summarized as follows:

**1)** *Prohibitive computational costs.* Existing methods are constrained to grounding objects within a well-matched local 3D scene, whereas CSSRG requires the traversal of numerous scenes, inevitably resulting in an inescapable crisis in both efficiency and performance. When 3DVG methods (*i.e.*, VisTA [10]) are extended from a single local scene to reasoning across 141 ScanRefer rooms [2], the inference time catastrophically increases by nearly 75 times, while performance declines to only 11.7% of the original.

**2)** *Complex cross-modal spatial alignment.* In contrast to 3DVG, which presumes a predefined text-scene correspondence, CSSRG requires achieving more challenging cross-modal alignment of the query texts with both relevant scenes and target objects. However, existing brief object descriptions fail to align comprehensively with the relevant complex scenes (*i.e.*, weakly correlated positives) and may resemble local parts of unrelated scenes (*i.e.*, false negatives), called the partial alignment problem. For example, a description text aiming to locate a *bed* typically focuses only on the *bed*'s attributes and its placement, while ignoring other scene-level salient information, leading to the partial alignment problem [38, 39].

# 4 Baseline: CoRe

Give a dataset $\mathcal{D} = \{\mathcal{T}, \mathcal{S}\}$, where $\mathcal{T} = \{X_i^t\}_{i=1}^M$ and $\mathcal{S} = \{X_j^s\}_{j=1}^N$ are the text and 3D scene sets with $M$ and $N$ samples, respectively. CSSRG task requires establishing the positive correspondence (*i.e.*, , $y_{ij} = 1$, with the rest negative pairs $y_{i\cdot} = 0$) between text $X_i^t$ and corresponding scene $X_j^s$, and grounding the target object $y_i^t$.

To tackle this challenging task, we propose a novel Cross-Scene 3D Object Reasoning Framework (CoRe), as illustrated in Figure 4. To be specific, our CoRe incorporates an innovative Robust Text-Scene Aligning module (RTSA) and Tailored Word-Object Associating module (TWOA), realizing a matching-then-grounding pipeline to handle the task-specific challenges. The model can be optimized via gradient descent based on the overall objective function of the batch, as shown below:

$$\mathcal{L} = \mathcal{L}_c + \lambda_m \mathcal{L}_m + \lambda_g \mathcal{L}_g, \tag{1}$$

where $\mathcal{L}_c$ is the loss for object semantic aligning in CoRe, $\mathcal{L}_m$ and $\mathcal{L}_g$ are the loss terms employed by the RTSA and TWOA, $\lambda_m$ and $\lambda_g$ are the trade-off parameters, respectively. In the following sections, we will elaborate on the framework and two novel modules of CoRe.

## 4.1 Cross-Scene 3D Object Reasoning Framework

In CoRe, we innovatively introduce a matching-then-grounding pipeline for CSSRG. More specifically, we first perform feature encoding. On the one hand, the text $X_i^t$ is encoded into $d$-dimension features $Z_i^w \in \mathbb{R}^{M_i \times d}$ via a pre-trained BERT [40], where $M_i$ is the number of words in $i$-th text. On the other hand, the collection of scene objects $X_j^s$ segmented through the pre-trained Mask3D [41] is encoded by the pre-trained PointNet++ [42], obtaining the object embedding set $\bar{Z}_j^o$. Then, these

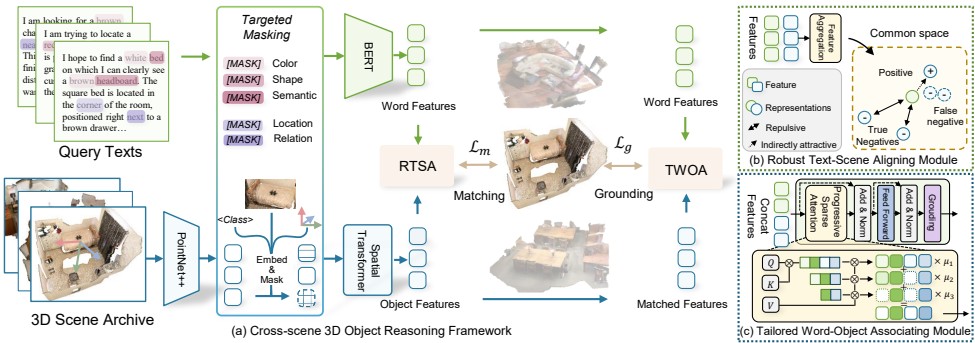

Figure 4: The pipeline of CoRe. First, modality-specific networks extract fine-grained features with a targeted masking strategy applied in each modality. Second, a Robust Text-Scene Aligning module (RTSA) adaptively aggregates multimodal features into common representations and ensures robust text-scene matching by only using negative pairs. Finally, a Tailored Word-Object Associating module (TWOA) adopts a Screening Attention mechanism for word-object association in a stepwise manner, performing object grounding. $\mathcal{L}_m$ and $\mathcal{L}_g$ are the losses employed by RTSA and TWOA.

embeddings are fed into Spatial Transformer [43] with spatial relation and class semantics, obtaining object features $Z_j^o \in \mathbb{R}^{N_j \times d}$, where $N_j$ is the number of objects in $j$-th scene.

To enhance our CoRe's perception within 3D scenes, we introduced a targeted masking strategy during encoding, focusing on visual attributes (*e.g.*, class semantics, color, shape) and spatial details. Specifically, for text modality, we mask the original text by replacing words in predefined attributive and spatial vocabularies with *[MASK]* at a specified probability. For point-cloud modality, we selectively mask the object, class-semantic, and positional embeddings.

After feature encoding, the fine-grained features of two modalities are aggregated into common representations for text-scene matching in our RTSA, which aligns description texts with the corresponding scenes. TWOA performs feature association between words and objects only within the matched scenes, which obtains fused features $\tilde{Z}_j^o \in \mathbb{R}^{N_j \times d}$ for efficient object reasoning. To maintain the discrimination of objects, we follow VisTA [10] to bridge the feature space and semantic space of the objects throughout the pipeline. This could be formulated as:

$$\mathcal{L}_c = \frac{1}{K} \sum_{j=1}^{K} \left( \mathcal{H}(\bar{\boldsymbol{p}}_j^o; \boldsymbol{y}_j^o) + \mathcal{H}(\boldsymbol{p}_j^o; \boldsymbol{y}_j^o) + \mathcal{H}(\tilde{\boldsymbol{p}}_j^o; \boldsymbol{y}_j^o) \right), \tag{2}$$

where $\mathcal{H}$ is Cross Entropy (CE), $K$ is size of the mini-batch, $\boldsymbol{y}_j^o$ is class label of the $j$-th scene objects corresponding to $i$-th text, $\bar{\boldsymbol{p}}_j^o, \boldsymbol{p}_j^o, \tilde{\boldsymbol{p}}_j^o$ are class semantic predictions of $\bar{Z}_j^o, Z_j^o, \tilde{Z}_j^o$.

## 4.2 Robust Text-Scene Aligning module

To facilitate text-scene Cross-Modal Matching (CMM), following [44], we first aggregate fine-grained object and word features into global text and scene representations with two different focuses: one emphasizes discriminative tokens (*i.e.*, words and objects), and the other focuses on informative feature dimensions. To effectively leverage their complementary focuses, we adaptively combine them, ultimately obtaining common representations of two modalities (*i.e.*, $\boldsymbol{z}_i^t$ and $\boldsymbol{z}_j^s$).

After obtaining the representations, we try to establish cross-modal alignment in the common space to facilitate matching from text to scene. To tackle the partial alignment between scenes and texts in CSSRG, we first employ the complementary learning paradigm [45] with GCE [46] expansion for a robust solution, as shown below:

$$\mathcal{L}_m = \frac{1}{K} \sum_{i,j}^{K} (1 - y_{ij}) \left( \frac{1 - (1 - \overrightarrow{s_{ij}})^q}{q} + \frac{1 - (1 - \overleftarrow{s_{ij}})^q}{q} \right), \tag{3}$$

where $\overrightarrow{s_{ij}} = \frac{\exp(\boldsymbol{z}_i^{t\top} \boldsymbol{z}_j^s / \tau)}{\sum_k^K \exp(\boldsymbol{z}_i^{t\top} \boldsymbol{z}_k^s / \tau)}, \overleftarrow{s_{ij}} = \frac{\exp(\boldsymbol{z}_i^{s\top} \boldsymbol{z}_j^t / \tau)}{\sum_k^K \exp(\boldsymbol{z}_i^{s\top} \boldsymbol{z}_k^t / \tau)}$ is the similarity between the $i$-th scene/text feature and the $j$-th text/scene feature, $q$ is a hyper-parameter, $\tau \in (0, 1]$ is the temperature parameter. Minimizing the Equation (3) would reduce the similarity of negative pairs, embracing discrimination without employing partially aligned positive pairs that prone to be noisy.

Subsequently, to mitigate the impact of false-negative pairs, we set the fixed $q$ as a variable that adaptively controls the loss robustness for each pair. Specifically, we aim to associate the loss robustness with the reliability of negative pairs, enhancing the robustness of the loss for unreliable pairs while preserving discrimination for reliable ones. In simple terms, we empirically set $q = \overleftarrow{s_{ij}}$, where the similarity of pairs serves as a proxy for their reliability, with pairs exhibiting higher similarity being prone to constitute false negatives. For more aggregation details of the feature aggregation and the analysis on proposed $\mathcal{L}_m$, please refer to our Supplementary Material.

### 4.3 Tailored Word-Object Associating module

Although RTSA has established robust text-scene alignment by aligning rich global representations, reasoning from the redundant context to the target objects requires precise finer-grained alignment. However, dense attention mechanisms (*e.g.*, Self-Attention [47], Cross-Attention), commonly used for fine-grained information fusion, force associations across all input features. It would make the model inevitably pay attention to numerous irrelevant features. In contrast, sparsity control in sparse attention mechanisms relies on prior knowledge, potentially leading to the loss of crucial information.

To address the issues, we propose a Tailored Word-Object Associating module (TWOA) incorporated with a novel Screening Attention mechanism (ScA), built on the Transformer encoder architecture. Specifically, it calculates attention across word and object features based on the *Query* and *Key*, then evenly divides the attention into $L$ segments based on the $L$-Quantile[48], from high to low. Assuming the $i$-th text matches $j$-th scene, it is written as:

$$A_i = \text{Sort}\left(W_q Z_i^c (W_k Z_i^c)^\top\right) = [A_{i1}; \cdots ; A_{iL}], \tag{4}$$

where $A_i \in \mathbb{R}^{(M_i+N_j)\times(M_i+N_j)}$ is the attention scores across the word-object features, $W_q$ and $W_k$ is projection matrices to map $Z_i^c$ to *Query* and *Key*, $Z_i^c = [Z_i^w; Z_j^o] \in \mathbb{R}^{(M_i+N_j)\times d}$ is the $i$-th word-object concatenated feature, and $A_{ik} \in \mathbb{R}^{(M_i+N_j)\times(M_i+N_j)/L}$ is the $k$-th attention segment of the $i$-th sample. Attention across segments varies in effectiveness, with top segments exhibiting greater weighting efficacy. Subsequently, we progressively screen the attention segments from low to high, resulting in $L$ distinct degrees of attention retention. We dynamically combine these attentions with the *Value* in a stepwise accumulation manner, enabling a progressive refinement of fine-grained feature associations through the flexible attention screening, which is written as:

$$\tilde{Z}_i^c = f_u(\sum_{j=1}^{L} \mu_j \tilde{A}_{ij}(W_v Z_i^c)^\top; \theta_u), \tag{5}$$

where $\tilde{Z}_i^c = [\tilde{Z}_i^w; \tilde{Z}_j^o]$ is the multimodal feature after applying ScA (*i.e.*, $\tilde{Z}_i^w$ and $\tilde{Z}_j^o$ are the word and object features, respectively), $\{\mu_j\}_{j=1}^{L}$ is the learnable coefficients, $\tilde{A}_{ij} = \text{Softmax}([A_{i1}; \cdots ; A_{ij}; 0; \cdots ; 0])$ is top-$j$ accumulative screened attention, where attention after the $j$-th segment is masked. $W_v$ is projection matrices to map $Z_i^c$ to *Value*, and $f_u(\cdot; \theta_u)$ represents the Transformer fusion function. With ScA, TWOA could gradually adjust the volume of features involved in the fine-grained association, filtering out excessive information and achieving precise fine-grained alignment between textual descriptions and target objects.

Finally, we can calculate the grounding scores of each object feature to infer the target object through a grounding layer with a weight matrix $W_g$, and we supervise it with CE loss:

$$\mathcal{L}_g = \frac{1}{K} \sum_{i}^{K} \mathcal{H}(W_g \tilde{Z}_i^o, y_i^t). \tag{6}$$

## 5 Dataset: CrossScene-RETR

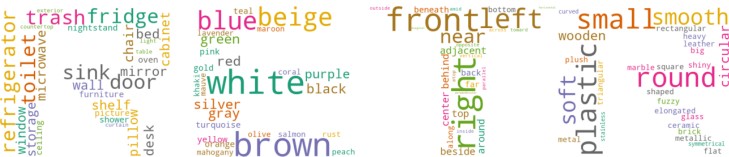

Figure 5: Word clouds of the proposed CrossScene-RETR dataset.

To the best of our knowledge, the descriptions in existing 3DVG datasets (*e.g.*, ScanRefer [2], *etc.*) provide fewer than 5 information points (*e.g.*, object class, colors, positions, *etc.*) and only cover 1.8 objects, as shown in Table 1. Such limited informational texts, when applied to CSSRG, undoubtedly intensify the challenges and complexity of the cross-modal spatial alignment.

To tackle the issues, we establish a discrimination benchmark dataset specific to CSSRG, namely CrossScene-RETR. In CrossScene-RETR, the 3D point-cloud data and object annotations are sourced from the widely used ScanNet dataset [49]. Query descriptions of objects with cross-scene discrimination are generated through our spatial analysis texts of scenes and corresponding corpora from existing text datasets (*i.e.*, ScanRefer [2], Nr3D [50], Sr3D [50], and ScanQA [51]). We will elaborate on its construction and statistics in the following.

## 5.1 Dataset Construction

We establish description texts of our CrossScene-RETR in four phases, with the pipeline shown in Figure 6.

**1)** *Scene Analysis Phase*: We first conduct an intra-scene analysis to assess object discrimination and localization within scenes. In addition, inter-scene analysis is performed to determine whether similar objects frequently appear across various scenes. Based on these, we gather extensive spatial information about objects and divide objects into *conspicuous*, *regular*, and *confusing* subsets to reflect the challenge level of each object in CSSRG, guiding text generation and model evaluation.

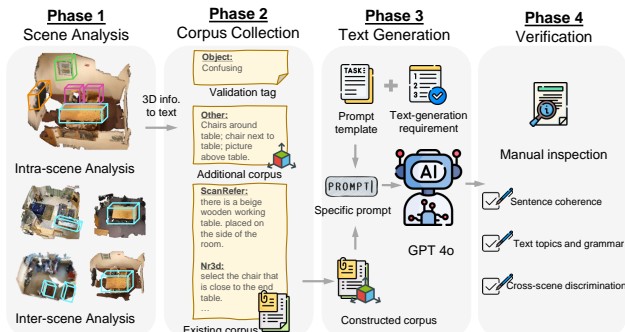

Figure 6: Construction pipeline of CrossScene-RETR.

**2)** *Corpus Collection Phase*: We gather available descriptions from ScanRefer, Nr3D, Sr3D, and ScanQA related to every object, covering attributes such as color, shape, position information, *etc*. Additionally, we enriched these preliminary texts on object positions and relative spatial relationships based on the *Scene Analysis Phase*. Based on these, we construct a rich corpus for scene objects.

**3)** *Text Generation Phase*: We utilize GPT-4o as the generation model and design a prompt template tailored to the task requirements. To ensure relevance for real-world applications, we switch four generation requirements in the template to generate four style description subsets: *characteristic-focused*, *spatial-information-focused*, *comprehensive*, and *fuzzy* subsets.

**4)** *Verification Phase*: We manually assess the generated descriptions for linguistic coherence and grammatical accuracy. In addition, we employ several staff to remove erroneous descriptions and eliminate ambiguous descriptions, ensuring the cross-scene discrimination of descriptions.

## 5.2 Brief Statistics

To comprehensively understand our proposed CrossScene-RETR, we provide a statistical comparison of it, as shown in Table 1. The results show that our descriptions are richer and more discriminative, which ensures that they are unambiguous for CSSRG applications. Due to space limitations, more comprehensive construction details and statistical analysis of CrossScene-RETR are provided in our Supplementary Material.

Table 1: Statistics comparison among four datasets.

| | Statistical indicators | ScanRefer | Nr3D | Sr3D | RETR |
|---|---|---|---|---|---|
| Overall | Average length | 17.9 | 11.4 | 9.7 | 77.7 |
| | Number of samples | 46,173 | 41,503 | 83,572 | 39,526 |
| | Vocabulary size | 6,919 | 6,951 | 196 | 22,485 |
| | Free-form | √ | √ | × | √ |
| Richness of description | Number of objects per text | 1.8 | 1.7 | 1.8 | 11.4 |
| | Number of characteristics per text | 1.5 | 1.6 | 0.0 | 5.9 |
| | Number of Spatial info. per text | 1.2 | 1.1 | 0.6 | 6.3 |
| | Number of info. points per text | 4.5 | 4.4 | 2.4 | 23.2 |
| | Text with Spatial info. (%) | 69.8 | 47.5 | 55.6 | 97.5 |
| | Text with color description (%) | 58.2 | 29.7 | 0.0 | 81.2 |
| | Text with shape description (%) | 20.3 | 6.5 | 0.0 | 45.0 |
| | Text with material description (%) | 13.0 | 2.1 | 0.7 | 38.5 |

Table 4: Scene matching and Object Grounding (OG) performance comparison on four datasets in terms of R@1, R@5, R@10, and Acc@0.25. The top of the table shows the results of fine-grained CMM methods, and the bottom shows the coarse-grained methods results. The highest results are shown in **bold** and the second highest results are underlined.

| Method | ScanRefer | | | | Nr3D | | | | Sr3D | | | | CrossScene-RETR | | | | T |
|---|---|---|---|---|---|---|---|---|---|---|---|---|---|---|---|---|---|
| | Scene matching | | | OG | Scene matching | | | OG | Scene matching | | | OG | Scene matching | | | OG | |
| | R@1 | R@5 | R@10 | Acc@0.25 | R@1 | R@5 | R@10 | Acc@0.25 | R@1 | R@5 | R@10 | Acc@0.25 | R@1 | R@5 | R@10 | Acc@0.25 | |
| NAAF | 9.17 | 35.05 | 54.17 | 2.17 | 10.35 | 34.32 | 50.63 | 1.48 | 3.36 | 12.76 | 24.83 | 0.37 | 28.72 | 66.34 | 78.03 | 2.05 | 58 ms |
| CHAN | 10.14 | 38.03 | 55.35 | 3.15 | 14.03 | 40.06 | 59.42 | 2.06 | 5.36 | 21.34 | 34.78 | 1.04 | 30.35 | 67.51 | 78.35 | 4.01 | 63 ms |
| CRCL-F | 10.05 | 37.61 | 56.13 | 2.08 | 13.95 | 41.42 | 59.50 | 1.88 | 5.34 | 24.16 | 35.44 | 1.13 | 28.35 | 65.45 | 78.17 | 3.47 | 63 ms |
| VSE∞ | 9.32 | 37.53 | 55.78 | - | 12.73 | 38.56 | 56.77 | - | 5.71 | 21.99 | 35.29 | - | 30.31 | 67.07 | 77.59 | - | 13 ms |
| HREM | 9.13 | 38.35 | 54.38 | - | 12.81 | 39.86 | 57.31 | - | 5.42 | 21.92 | 33.57 | - | 29.13 | 66.16 | 78.73 | - | 18 ms |
| ESA | 10.13 | 37.41 | 55.63 | - | 13.74 | 39.25 | 58.74 | - | 5.52 | 21.25 | 34.44 | - | 29.24 | 67.08 | 79.78 | - | 18 ms |
| CRCL-C | 10.78 | 38.05 | 54.29 | - | 13.08 | 39.90 | 58.69 | - | 4.94 | 19.49 | 31.90 | - | 28.31 | 65.63 | 76.97 | - | 14 ms |
| Ours | **13.29** | **38.84** | **56.20** | **6.24** | **14.56** | **43.54** | 58.23 | **5.86** | **5.95** | 23.22 | 34.79 | **3.52** | **36.48** | **68.53** | **81.44** | **22.99** | 54 ms |

# 6 Experiments

In the experiments, we adopt the 3D set along with CrossScene-RETR, ScanRefer, Nr3D, and Sr3D text sets for CSSRG evaluation. We compare our CoRe with 13 state-of-the-art methods, which include: 3DVG methods (*i.e.*, ScanRefer [2], 3D-BUTD [52], EDA [53], VisTA [10], GPS [3], and TSP3D [54]) and CMM methods (*i.e.*, NAAF [32], CHAN [55], VSE∞ [33], HREM [56], ESA [57], coarse-grained CRCL-C and fine-grained CRCL-F [58]). In addition, we integrate advanced CMM and 3DVG methods to construct

Table 2: Performance comparison on CrossScene-RETR in terms of Acc@0.25 (0.25) and Acc@0.5 (0.5). The highest results are shown in **bold** and the second highest are underlined.

| Method | Conspicuous | | Regular | | Confusing | | Overall | | T |
|---|---|---|---|---|---|---|---|---|---|
| | 0.25 | 0.5 | 0.25 | 0.5 | 0.25 | 0.5 | 0.25 | 0.5 | |
| CRCL-F | 2.35 | 2.35 | 2.54 | 2.22 | 2.69 | 2.58 | 2.54 | 2.34 | 63 ms |
| Vista | 7.06 | 4.57 | 8.70 | 6.26 | 7.38 | 5.08 | 8.04 | 5.62 | 14.2 s |
| GPS | 5.54 | 5.31 | 5.30 | 5.27 | 5.27 | 5.17 | 5.34 | 5.24 | 27.2 s |
| TSP3D | 4.85 | 4.36 | 4.61 | 4.21 | 4.32 | 3.74 | 4.58 | 4.12 | 14.3 s |
| HREM+VisTA | 20.41 | 19.46 | 16.88 | 15.43 | 14.13 | 12.88 | 17.25 | 15.96 | (18 + 45) ms |
| ESA+GPS | 21.26 | 20.08 | 14.34 | 13.18 | 13.98 | 12.82 | 15.60 | 14.44 | (18 + 171) ms |
| Ours | **28.67** | **26.52** | **22.45** | **20.82** | **19.83** | **18.20** | **22.99** | **21.26** | 54 ms |

matching-then-grounding baselines (*i.e.*, HREM+VisTA, ESA+GPS) for comparison. We report the following metrics for CSSRG evaluation: **1)** Acc@$k$ ($k \in \{0.25, 0.5\}$): The reasoning accuracy which requires matching the correct scene while the predicted box overlaps the ground truth with IoU $> k$. **2)** R@$K$ ($K \in \{1, 5, 10\}$): Scene retrieval recall at $K$, following the CMM metrics [21]. **3)** T: Average inference time per query. Due to space limitations, the introduction to the adopted datasets, implementation details of the methods, could be found in our Supplementary Material.

## 6.1 Comparison with the State-of-the-Arts

The performance comparison results between CoRe, 3DVG, and constructed matching-then-grounding methods are reported in Tables 2 and 3, and the scene matching comparison results between our CoRe and the CMM methods are presented in Table 4. These results could yield the following observations: **1)** Compared to the inferior performance in existing datasets, the performance in our CrossScene-RETR shows a substantial improvement. This demonstrates that it encapsulates greater discrimination, addressing the complex cross-modal spatial alignment chal-

Table 3: Performance comparison on ScanRefer, Nr3D, and Sr3D using Acc@0.25/0.5 (0.25/0.5). Best results are in **bold**, second-best are underlined.

| Method | ScanRefer | | | | | | Nr3D | Sr3D | T |
|---|---|---|---|---|---|---|---|---|---|
| | Unique | | Multiple | | Overall | | Overall | Overall | |
| | 0.25 | 0.5 | 0.25 | 0.5 | 0.25 | 0.5 | 0.25 | 0.25 | |
| 3D-BUTD | 6.14 | 2.90 | 2.17 | 2.03 | 2.90 | 2.19 | 0.00 | 0.20 | 13.2 s |
| EDA | 3.31 | 0.00 | 0.40 | 0.00 | 0.70 | 0.00 | 2.50 | 0.66 | 23.1 s |
| VisTA | 5.63 | 5.44 | 5.16 | 5.34 | 5.23 | 5.36 | 0.63 | 0.25 | 12.1 s |
| GPS | 5.04 | 4.11 | 3.84 | 3.80 | 4.04 | 3.85 | 0.20 | 0.35 | 21.2 s |
| TSP3D | 3.17 | 2.82 | 1.53 | 1.32 | 1.78 | 1.55 | 1.14 | - | 14.3 s |
| HREM+VisTA | 6.14 | 5.34 | 4.74 | 4.14 | 5.00 | 4.36 | 4.57 | 2.14 | (18 + 42) ms |
| ESA+GPS | 6.60 | 5.86 | 5.21 | 4.91 | 5.43 | 5.06 | 4.78 | 2.97 | (18 + 164) ms |
| Ours | **7.39** | **6.88** | **5.98** | **5.54** | **6.24** | **5.79** | **5.86** | **3.52** | 51 ms |

lenge. **2)** Our CoRe achieves inference efficiency comparable to CMM methods, outperforming 3DVG methods by a factor of 250. This validates the effectiveness of our two-stage framework for CSSRG. **3)** Our CoRe achieves better results both in scene matching and in object grounding compared with baselines, showing its superiority in overcoming the specific challenges. **4)** The performance on CSSRG is relatively low, indicating that the methods still face difficulties in handling CSSRG, and call for more advanced solutions.

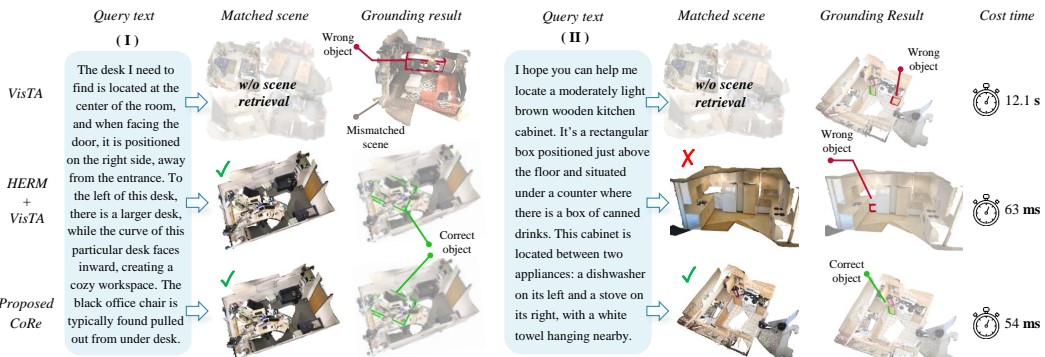

Figure 7: Some CSSRG instances on CrossScene-RETR among VisTA, HREM+VisTA, and CoRe. Correctly matched scenes are marked with a **green** tick, otherwise the **red** cross. Correctly located objects are highlighted with **green** boxes, otherwise the **red** boxes.

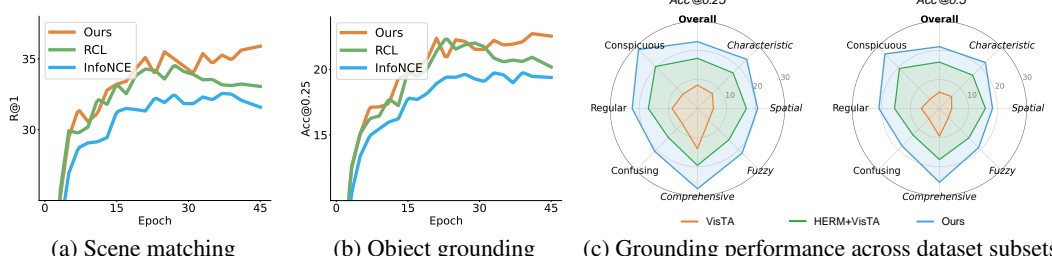

(a) Scene matching      (b) Object grounding      (c) Grounding performance across dataset subsets

Figure 8: (a) and (b) show matching and grounding performance comparison of our CoRe and its variants using contrastive learning (InfoNCE) and complementary learning (RCL), evaluated by R@1 and Acc@0.25. (c) shows object grounding performance of CoRe across different subsets of CrossScene-RETR, in terms of Acc@0.25 and Acc@0.5. Overall performance is highlighted in **bold**, and subsets with varying challenge levels and description styles are shown in *italics* and normal font.

## 6.2 Ablation Study

In this section, we investigate the contribution of each adopted component to CSSRG. For a comprehensive comparison, we ablate or substitute each component and conduct the variants with the same experimental setting on the CrossScene-RETR dataset. Specifically, we ablate the Targeted Masking and RTSA from our framework. In addition, we replace the matching loss with its fixed $q$ variant (*i.e.*, $\mathcal{L}_m^{\bar{q}}$) in the RTSA, and the ScA with dense Self-/Cross-Attention (SA/CA) in our TWOA. The results in Table 5 lead to the following observation: **1)** Removing or replacing any

Table 5: Ablation studies for components of our CoRe on CrossScene-RETR. R@Sum is the sum of R@1, R@5, R@10. ✓ stands for use.

| Mask | RTSA | TWOA | R@Sum | Acc@0.25 | Acc@0.5 |
|------|------|------|-------|----------|---------|
| ✓ | ✓ | ✓ | 186.45 | 22.99 | 21.26 |
|  | ✓ | ✓ | 183.26 | 21.37 | 20.29 |
| ✓ |  | ✓ | 5.25 | 1.03 | 0.72 |
| ✓ | $\mathcal{L}_m^{\bar{q}}$ | ✓ | 177.94 | 21.18 | 20.29 |
| ✓ | ✓ | SA | 185.72 | 19.76 | 18.38 |
| ✓ | ✓ | CA | 177.43 | 17.53 | 16.47 |

component from CoRe results in performance degradation, highlighting the contribution of each component. **2)** The performance brought by the matching loss we use in RTSA is superior to the vanilla alternatives, demonstrating its contribution to robust text-scene matching in CSSRG. **3)** The replacement with dense attention degrades performance, proving that ScA alleviates the impact of redundant information on word-object association.

## 6.3 Visualization Analysis

To provide a comprehensive understanding of our proposed CrossScene-RETR dataset and CoRe baseline, we conduct visualization experiments. Specifically, we visualize the CSSRG result instances of VisTA, HREM+VisTA, and our CoRe, as shown in Figure 7. Additionally, we present the CSSRG performance comparisons between VisTA, HREM+VisTA and CoRe within varying challenge levels and description styles subsets in the proposed CrossScene-RETR. We present the performance comparison in scene matching and object grounding between CoRe and its variants

based on contrastive and complementary learning. Both are presented together in Figure 8. The following observations can be drawn from the results: **1)** Our CoRe achieves more accurate results and reasoning efficiency, demonstrating its ability to address the specific challenges of CSSRG. **2)** Diverse descriptions and scene objects in CrossScene-RETR pose various practical challenges to methods. Experimental results demonstrate that our CoRe achieves an extensive understanding and superior grounding performance of diverse texts and objects. **3)** Throughout the learning process, it is evident that non-robust variants exhibit suboptimal matching and grounding performance compared to our CoRe, highlighting the ability to mitigate partial text-scene alignment issue of CoRe.

## 7 Conclusion

In this paper, we introduce a new task, Cross-Scene Spatial Reasoning and Grounding (CSSRG), which extends 3D visual grounding to a broader, more practical, and more complex setting. To effectively tackle this task, we propose the **Cro**ss-Scene 3D Object **Re**asoning Framework (CoRe), integrating two novel modules: the Robust Text-Scene Aligning module (RTSA) and the Tailored Word-Object Associating module (TWOA). Specifically, CoRe adopts a matching-then-grounding pipeline, enabling efficient cross-scene grounding. RTSA mitigates the issue of partial alignment by refining text-scene association, while TWOA enhances non-redundant word-object association, improving object grounding precision. Additionally, we introduce the CrossScene-RETR dataset, designed to evaluate the challenges of CSSRG more effectively. Extensive experiments on four datasets demonstrate the superiority and effectiveness of our CoRe, highlighting its potential for advancing cross-scene 3D reasoning and multimodal understanding.

## Acknowledgments

This work was supported in part by NSFC under Grant 62472295, 62176171, 62372315, U24B20174; in part by the National Key R&D Program of China under Grant 2024YFB4710604; in part by the Fundamental Research Funds for the Central Universities under Grant CJ202303, CJ202403; in part by Sichuan Science and Technology Planning Project under Grant 24NSFTD0130, 2024ZDZX0004, 2024NSFTD0049; in part by the Chengdu Science and Technology Project under Grant 2023-XT00-00004-GX; in part by System of Systems and Artificial Intelligence Laboratory pioneer fund grant under Grant HLJGGG20240327517-15, and in part by TCL science and technology innovation fund.

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
