# OpenReview forum: "Robust Cross-modal Alignment Learning for Cross-Scene Spatial Reasoning and Grounding"
_NeurIPS.cc/2025/Conference — NeurIPS 2025 poster_

### Official Review · Reviewer_41F3 · 2025-06-04

**Clarity:** 3
**Significance:** 3
**Originality:** 3
**Rating:** 5
**Confidence:** 2

**Summary:**

Targeting at locating a described object anywhere across an entire collection of 3D scenes, this paper proposes a Cross-Scene 3D Object Reasoning Framework to realise scene matching and object grounding. This paper also organises a new dataset. Experimental results verify the effectiveness of the proposed RTSA and TWOA modules.

**Questions:**

1. The author's method focuses on the accurate alignment of text and objects. Are there any more innovative designs in the perception and components of 3D environments?
2. Why does the proposed method not work well in the Sr3D dataset? More explanation should be provided.
3. The authors should give more explanations about the training process of the proposed method.
4. How about the running speed and model parameters compared with other methods?
5. For Tab. 1, how to get these statistics？ Are they the statistics provided in the documentation of those datasets or the authors' own rough estimates?
6. For the targeted masking strategy, how to decide the mask location and probability? Whether the choice of mask have a significant impact on the experimental results?

**Ethical Concerns:**

["NO or VERY MINOR ethics concerns only"]

**Final Justification:**

My concerns have been addressed in the rebuttal. I keep with my positive score. I believe that the contributions and presentation of the paper have met the requirements of the conference.

**Limitations:**

Yes. I think this paper has no potential negative societal impact.

**Quality:**

3

**Strengths And Weaknesses:**

Strengths:
1. This paper is generally well-written and easy to follow.
2. This paper effectively articulates the issues associated with cross-modal alignment, providing a clear and comprehensive explanation from my point.

Weaknesses:
1. The visual results are not clear. I cannot directly observe the advantages of the proposed method without the aid of the word descriptions, such as Fig. 6.
2. I think the Eq. (1) can be moved to the end of Section 4 when all the modules have been introduced, which can make the organisation more clear.
3. More details should be provided, such as the parameter settings of the loss function.
4. Some texts in the figures are not too small, such as Fig. 5.
5. A limitation part can be included in the paper to make this research more comprehensive.

---

> ### Author Rebuttal · Authors · 2025-07-31
>
> Thanks for your valuable comments and insightful suggestions. We have carefully looked into all the comments and suggestions. Attached is our point-by-point response.
>
> **Q1: The visual results are not clear. I cannot directly observe the advantages of the proposed method without the aid of the word descriptions, such as Figure 6.**
>
> **R1:** Thank you for pointing this out. **Figure 6** presents a visual comparison of the predictions from our method and two representative baselines under the matching-then-grounding pipeline, with many details noted in the caption. In the next version, we will enhance the figure by adding key annotations to the visual pipeline and using more prominent colors to highlight the advantages of the proposed method compared to existing baselines.
>
> **Q2: Eq. (1) can be moved to the end of Section 4 when all the modules have been introduced, making the organisation clearer.**
>
> **R2:** Thank you for your constructive suggestions regarding the organization and readability of our paper. In the next version, we will revisit the structure and adjust the placement of **Eq. (1)** in accordance with your recommendation.
>
> **Q3: More details should be provided, such as the parameter settings of the loss function.**
>
> **R3:** We have conducted a comprehensive sensitivity analysis of all parameters, as presented in **Section C.2 of our Supplementary Material**. Additionally, the parameter settings used in our implementation have been made publicly available and are detailed in **Section B.3 of our Supplementary Material**.
>
> **Q4: Some texts in the figures are not too small, such as Figure 5.**
>
> **R4:** Thank you for pointing that out. We will increase the figure size and text font in the next version.
>
> **Q5: A limitation part can be included in the paper to make this research more comprehensive.**
>
> **R5:** We have conducted discussions of the limitations and potential impact of our proposed method and data, which is included in **Section D of our Supplementary Material**.
>
> **Q6: The author's method (i.e., CORE) focuses on the accurate alignment of text and objects. Are there any more innovative designs in the perception and components of 3D environments?**
>
> **R6:** Yes. Our CORE includes innovative designs tailored for 3D environmental perception within both the Cross-Scene 3D Object Reasoning framework and the two proposed modules, namely the Robust Text-Scene Aligning (RTSA) module and the Tailored Word-Object Associating (TWOA) module. Specifically, our framework introduces a targeted masking strategy during the feature encoding stage, which masks objects, their class semantics, or their positional embeddings in the 3D environment. This targeted enhancement helps improve the model’s perception of object characteristics, semantics, and spatial positions within scenes. Secondly, the RTSA module aggregates global text and scene features by computing attention at both the feature and token levels for fine-grained word and object features. This enables the model to adaptively learn the interrelations among features (e.g., object color, shape, etc.) and the relative importance of objects within a scene, thereby enhancing scene perception. Finally, the TWOA module employs a novel Screening Attention mechanism to filter out features of objects and words that are irrelevant to the target object, thereby facilitating focused scene perception for precise grounding.
>
> **Q7: Why does the proposed method not work well in the Sr3D dataset?**
>
> **R7:** In fact, this issue stems from limitations of the Sr3D dataset itself. All existing methods, not just ours, generally exhibit poor performance on the Cross-Scene Spatial Reasoning and Grounding (CSSRG) task when evaluated on Sr3D. Specifically, CSSRG requires identifying a target object based on a text query across hundreds of candidate scenes. The scene-level discrimination of the queries has a direct impact on the performance, followed by grounding within the matched scene. However, in Sr3D, object descriptions across different scenes tend to be similar and lack sufficient scene-level discrimination, leading to significant performance degradation. This is also the motivation behind our introduction of the CrossScene-RETR dataset, which is designed to address the aforementioned limitation for a more reasonable and comprehensive evaluation of the CSSRG task.
>
> **Q8: The authors should give more explanations about the training process of the proposed method.**
>
> **R8:** Due to page limits, this part has been included in the Supplementary Material. Specifically, additional explanations on the two proposed modules and training details are provided in **Section B of our Supplementary Material**. Furthermore, the detailed training process of the method can be found in **Algorithm 1 on page 7 of our Supplementary Material**.
>
> **Q9: How about the running speed and model parameters compared with other methods?**
>
> **R9:** In **Table 2** and **Table 3** of our Experiment Section, we provide a detailed comparison of the running speed between our proposed method and existing 3D Visual Grounding (3DVG) methods. The results show that our CORE achieves **over a 100-fold increase** in average speed compared to the 3DVG methods, demonstrating its high efficiency in the CSSRG task. Additionally, our proposed CORE is a lightweight CSSRG baseline with only **201.92 M parameters**. Its parameter count is comparable to, or even smaller than, the recent 3DVG methods (e.g., VisTA, GPS, etc.), but slightly higher than that of earlier methods such a 3D-BUTD. Although these 3DVG methods have a comparable number of parameters, they struggle to achieve satisfactory efficiency on the CSSRG task, which further validates the advantage of our CORE’s matching-then-grounding pipeline for this challenging task.
>
> **Q10: For Table 1, how to get these statistics？ Are they the statistics provided in the documentation of those datasets or the authors' own rough estimates?**
>
> **R10:** The data in Table 1 are accurately computed based on our proposed CrossScene-RETR dataset. Additional statistics, visualizations, case studies, and analyses for our CrossScene-RETR and its subsets are provided in **Section A**, **Figures 1 to 4 and 8 to 10**, and **Table 1 of our Supplementary Material**. These materials help readers better understand and apply our proposed task and dataset.
>
> **Q11: For the targeted masking strategy, how to decide the mask location and probability? Whether the choice of mask have a significant impact on the experimental results?**
>
> **R11:** **1)** As mentioned in the paper, our masking strategy focuses solely on visual attributes such as class semantics, color, shape, and spatial details. Specifically, for text modality, we mask the original text by replacing words in predefined attributive and spatial vocabularies. For point-cloud modality, we selectively mask the object, class-semantic, and positional embeddings. It is worth noting that each targeted location has a **10%** probability of being masked. This value is relatively low and follows a similar setting to BERT. **2)** No. To thoroughly analyze its importance and sensitivity, we conducted a comparison on the CSSRG task using different masking probabilities $p$ ranging from **0 to 0.3** on the CrossScene-RETR dataset, as shown in the table below. The results indicate that a small degree of targeted masking consistently improves the performance, while excessive masking leads to weak performance degradation. This demonstrates that, without excessively masking information, the choice of mask does not significantly impact the experimental results. This experiment and analysis will be included in the next version of our paper.
>
> | $p$      | 0     | 0.05  | 0.1   | 0.15  | 0.2   | 0.3   |
> |----------|-------|-------|-------|-------|-------|-------|
> | Acc\@0.25 | 21.73 | 21.85 | 22.99 | 22.81 | 21.40 | 20.13 |

---

### Official Review · Reviewer_Esws · 2025-06-30

**Clarity:** 3
**Significance:** 3
**Originality:** 4
**Rating:** 5
**Confidence:** 5

**Summary:**

This paper introduces a practical and general task called Cross-Scene Spatial Reasoning and Grounding (CSSRG). The authors conduct a thorough analysis of the task and its associated challenges. To address these challenges, they propose a solution framework named CORE, along with a special dataset called CrossScene-RETR. Specifically, CORE consists of two modules that are designed to tackle the respective challenges of CSSRG, mitigating both computational overhead and the risks of cross-modal misalignment. Experiments provide baseline results for CSSRG and demonstrate the superiority of the proposed CORE.

**Questions:**

Why CORE efficiency is significantly higher compared to 3D visual grounding methods? A more comprehensive descriptions of the method and data subset partitioning should be declared. See the weakness section for more details.

**Ethical Concerns:**

["NO or VERY MINOR ethics concerns only"]

**Final Justification:**

Thank you for the author's patient response. My concerns have been well addressed, and I recommend accepting this work.

**Limitations:**

yes

**Paper Formatting Concerns:**

This paper has no major formatting issues.

**Quality:**

4

**Strengths And Weaknesses:**

Strengths:
1.This paper introduces a new task and proposes a novel method and dataset for it, along with baseline results and detailed analysis. I believe the overall effort involved is substantial.

2. The paper is well-motivated and easy to follow. The CSSRG task appears to be both more challenging and practical, and it is thoroughly analyzed by the authors.

3. The paper is well-structured, with particularly intuitive and informative figures that aid understanding.

4. The proposed method is shown to be effective through both the methodological design and experimental results. The underlying theoretical framework seems to be solid.

5. The dataset analysis presented in the paper is comprehensive, especially the data construction, statistics, and analysis in the supplementary material.

Weaknesses:
1.It appears that CORE also needs to iterate through all scenes to match them before grounding objects. Why its efficiency is significantly higher compared to 3D visual grounding methods?

2.Sec. 4 could benefit from a more detailed explanation, as certain aspects remain unclear. For instance, the process by which RTSA obtains common representations is not fully elaborated. Additionally, how does the variation in $q$ help mitigate the impact of false-negatives?

3.The use of data subsets is confusing. It seems there are two different strategies for grouping the data into subsets, yet Fig. 7 analyzes them together. It would be helpful to clarify the essential differences between these grouping methods.

---

> ### Author Rebuttal · Authors · 2025-07-31
>
> Thanks for your valuable comments and insightful suggestions. Attached is our point-by-point response.
>
> **Q1: It appears that CORE also needs to iterate through all scenes to match them before grounding objects. Why is its efficiency significantly higher compared to 3D Visual Grounding (3DVG) methods?**
>
> **R1:** Unlike 3DVG methods that exhaustively and sequentially traverse scenes, our proposed CORE efficiently 'traverses' all scenes in a retrieval-based manner to match the scene containing the target object. More specifically, in CORE, both the text queries and scenes are first aggregated into coarse-grained common representations. Then, supported by the proposed Robust Text-Scene Aligning (RTSA) module, the text queries can directly measure similarity with the 3D scenes in the common space. This process can be efficiently implemented via matrix multiplication on the GPUs. In contrast, conventional 3DVG methods, which only support single-scene inference, can only perform inference over one scene at a time. Each inference requires fine-grained reasoning between dozens of objects and words, which significantly undermines their efficiency in cross-scene scenarios.
>
>
> **Q2: Section 4 could benefit from a more detailed explanation, as certain aspects remain unclear. For instance, 1) the process by which RTSA obtains common representations is not fully elaborated. 2) How does the variation help mitigate the impact of false-negatives?**
>
> **R2:** Due to page limits, the details of **Section 4** have been fully elaborated in **Section B of our Supplementary Material**. More specifically, **1)** Details on how RTSA derives common representations are included in **Section B.1 and Figure 5 of our Supplementary Material**. Specifically, we design attention at the token-feature and feature-dimension level for fine-grained word and object features, enabling the model to learn adaptive interrelation-focused and feature-focused weights. Subsequently, the two different weights are applied to the fine-grained features, aggregating them into common representations. **2)** The variation $\mathcal{L}\_{m}$ introduces an adaptive parameter $q$, which adaptively increases for more reliable false negatives. As proven in **Section B.2 of our Supplementary Material**, as $q$ approaches 1 from 0, the robustness of $\mathcal{L}\_{m}$ gradually improves and converges to its lower bound, i.e., the MAE loss, which could prevent overfitting to false-negatives.
>
> **Q3: The use of data subsets is confusing. It seems there are two different strategies for grouping the data into subsets, yet Figure 7 analyzes them together.**
>
> **R3:** Sorry for the confusion. We independently divide our CrossScene-RETR into different subsets based on two factors: the challenge level of object retrieval and the linguistic styles of the query texts. The former follows the splitting criteria of ScanRefer [A], and is detailed in the Scene Analysis Phase in **Section 5.1**. The latter is described in the Text Generation Phase in **Section 5.1** and further elaborated in **Section A.2 of our Supplementary Material**. Each subset has its own characteristics and analytical significance. Therefore, we present them collectively in **Figure 7** using radar charts to provide a comprehensive comparison of baseline performances. In future versions, we will integrate and clarify the description of the subset division.
>
> **References:**
>
> [A] Chen, Dave Zhenyu, Angel X. Chang, and Matthias Nießner. "Scanrefer: 3d object localization in rgb-d scans using natural language." European conference on computer vision. Cham: Springer International Publishing, 2020.

---

### Official Review · Reviewer_Doy1 · 2025-07-02

**Clarity:** 3
**Significance:** 3
**Originality:** 3
**Rating:** 5
**Confidence:** 4

**Summary:**

In this paper, the authors aim to investigate a novel task named as Cross-Scene Spatial Reasoning and Grounding (CSSRG). Different from the existing 3D Visual Grounding (3DVG) task, CSSRG extend the perception of a described object to an entire collection of 3D scenes rather than the predefined scene in 3DVG. To address this challenge, the authors propose CoRe, which adopts a matching-then-grounding paradigm to effectively reduce computational costs. CoRe comprises a Robust Text-Scene Aligning module (RTSA) for robust scene matching and a Tailored Word-Object Associating module (TWOA) for object grounding. The experiments on various datasets (ScanRefer/Nr3D/Sr3D/RETR) demonstrate the superiority and effectiveness of CoRe compared with existing SOTAs.

**Questions:**

1. **[Clarification about Spatial Reasoning]**. Could the authors provide some clarification about the “Spatial Reasoning” part in the task of Cross-Scene Spatial Reasoning and Grounding (CSSRG)? As discussed in Weakness #2, I think the proposed method can only handle the grounding task. However the spatial reasoning is a much larger concept, rather than a single grounding task. More tasks should be involved if “Spatial Reasoning” is used in your paper.
 2. **[Number of Scenes]**. From Figure 2, 141 scenes are used in CSSRG task, whereas only 1 scene is used in 3DVG task. Does the number of scenes affect the performance of CSSRG?

**Ethical Concerns:**

["NO or VERY MINOR ethics concerns only"]

**Final Justification:**

The authors have solved my questions. I have no further concerns.

**Limitations:**

yes

**Quality:**

3

**Strengths And Weaknesses:**

**Strength:**
 1. This paper firstly investigates the Cross-Scene Spatial Reasoning and Grounding (CSSRG) task. Different from many existing works that concentrate on the 3DVG task with predetermined scene configuration, CSSRG can handle an entire collection of 3D scenes. It is more challenging and meaningful.
 2. The overall organization of this paper is clear and easy to follow.
 3. The proposed dataset (CrossScene-RETR) is of great significance and inspiring for the future works.

**Weakness:**
 1. The figures (Fig. 4, 5, 6) are hard to read. The text in these figures are too small.
 2. I am kind of confused about the “Spatial Reasoning” part in the task of Cross-Scene Spatial Reasoning and Grounding (CSSRG). From Figure 3, I can understand that the proposed pipeline first find the matching scene with the given prompt, and then conduct visual grounding to find the exact spatial location according to the given prompt. However, this process only includes the grounding part, the concept of spatial reasoning seems somewhat overly broad. Maybe more tasks should be involved rather than a single grounding task.
 3. The introduction of the proposed CrossScene-RETR dataset is too brief in Section 5. I have read the detailed introduction of the proposed dataset in the appendix. I think the proposed dataset is an important contribution to the community and should be highlighted in the manuscript.
 4. Failure case analysis is missing. The authors are suggested to provide a failure case analysis.

---

> ### Author Rebuttal · Authors · 2025-07-31
>
> Thanks for your valuable comments and insightful suggestions. Attached is our point-by-point response.
>
> **Q1: The figures (Figure 4, 5, 6) are hard to read. The texts in these figures are too small.**
>
> **R1:** Thank you for the suggestion! We will increase the figure size and text font in the next version.
>
> **Q2: 1) Could the authors provide some clarification about the “Spatial Reasoning” part in Cross-Scene Spatial Reasoning and Grounding (CSSRG)? CSSRG seems to only include the grounding stage, and the concept of spatial reasoning appears somewhat overly broad. 2) Maybe more tasks should be included instead of relying on a single grounding task.**
>
> **R2:** **1)** “Spatial Reasoning” runs through the entire CSSRG solution pipeline, not just the grounding stage. More specifically, during the scene matching stage, 3D scene features are reasoned over a collection of scenes based on text features to identify the scene whose spatial structure best matches the description. In the object grounding stage, reasoning is performed within the matched scene to ground the target object described in the text description. **2)** This paper is an initial attempt to perform spatial reasoning and object grounding across a scene collection. In the future, we will continue to enrich CSSRG downstream tasks based on your valuable suggestions.
>
> **Q3: The introduction of the proposed CrossScene-RETR dataset is too brief. This dataset is an important contribution to the community and should be highlighted in the manuscript.**
>
> **R3:** Thank you for recognizing the importance of our proposed dataset! Since we introduced a new task, CSSRG, it is essential to provide a clear and comprehensive description of it. Due to page limits, we had to condense the dataset and method sections, with much of the content provided in our Supplementary Material. In the future version, we will incorporate more details and insightful statistics from the Supplementary Materials into the main body.
>
> **Q4: Failure case analysis is missing.**
>
> **R4:** Thanks. In the next version, we will include some failure case analysis to provide a more comprehensive evaluation of our CORE method.
>
> **Q5: 1) From Figure 2, 141 scenes are used in the CSSRG task, whereas only 1 scene is used in the 3DVG task. 2) Does the number of scenes affect the performance of CSSRG?**
>
> **R5:** **1)** You may have misunderstood the implication of **Figure 2**. It illustrates that existing 3D visual grounding (3DVG) methods require **0.16 seconds per scene** to perform inference on the 3DVG task. When directly applied to the CSSRG task, which contains 141 scenes, its inference time dramatically increases to **21.2 seconds per scene**, accompanied by a significant performance degradation. Therefore, directly applying 3DVG methods to the CSSRG task is impractical, and adopting an efficient matching-then-grounding approach (i.e., our proposed CORE method) becomes necessary. **2)** As shown in **Figure 2** and discussed above, the number of scenes significantly affects the performance of conventional 3DVG methods on the CSSRG task. In contrast, CORE attempts to first efficiently align global scene and text representations and then perform object grounding, which is virtually unaffected by the number of scenes.

---

> > ### Comment · Reviewer_Doy1 · 2025-08-06
> >
> > The authors have solved my questions. I have no further concerns.

---

> > > ### Author Response · Authors · 2025-08-06
> > >
> > > Thank you for your response. I'm pleased to hear that your concerns have been fully addressed, and I sincerely appreciate your recommendation for acceptance.

---

### Official Review · Reviewer_aLPd · 2025-07-03

**Clarity:** 3
**Significance:** 2
**Originality:** 3
**Rating:** 4
**Confidence:** 3

**Summary:**

The paper introduces Cross-Scene Spatial Reasoning and Grounding (CSSRG), where an agent must find a language-described object anywhere in a large archive of 3D scenes. To tackle the prohibitive traversal cost and partial text-scene alignment, the authors propose CoRe, a two-stage “match-then-ground” framework combining a Robust Text-Scene Aligning module with a Screening-Attention-based Tailored Word-Object Associating module. They also plan to release the CrossScene-RETR dataset, whose richer descriptions sharpen evaluation, and show CoRe beats thirteen baselines while running ≈250× faster than single-scene methods.

**Questions:**

I would appreciate if you could address my questions regarding limitations.

**Ethical Concerns:**

["NO or VERY MINOR ethics concerns only"]

**Final Justification:**

After thoroughly reading the authors' rebuttal, I was able to resolve most of my concerns, especially both model architecture-wise and result-wise. Therefore, I have decided to raise my rating of borderline reject to borderline accept.

**Limitations:**

Yes

**Quality:**

3

**Strengths And Weaknesses:**

**Strengths**

* Frames a needed research problem (CSSRG) that reflects real agent memory search.
* Plans to release CrossScene-RETR, enabling future benchmarking.
* Two-stage CoRe cuts computation yet lifts accuracy over 13 baselines.

**Weaknesses**

* Despite gains, CSSRG accuracy stays low (≈23 % Acc\@0.25), so practical use is distant.
* Experiments rely on a single 3D backbone; generality to newer encoders is untested.
* The four 3D VG text datasets use heterogeneous text formats, yet the paper gives no preprocessing strategy for syncing them.
* Scope is limited to indoor static scenes; outdoor or dynamic settings remain unexplored.

---

> ### Author Rebuttal · Authors · 2025-07-31
>
> Thanks for your valuable comments and insightful suggestions. We have carefully looked into all the comments and suggestions. Attached is our point-by-point response.
>
> **Q1: The accuracy of Cross-Scene Spatial Reasoning and Grounding (CSSRG) stays low (≈23 % Acc\@0.25), so practical use is distant.**
>
> **R1:** CSSRG is a challenging task introduced for the first time in our paper, which requires reasoning across hundreds of scenes to locate **only one** single target object based solely on a text query. On the one hand, compared to existing baselines, which achieve less than **5%** CSSRG performance in terms of Acc\@0.25 on the CrossScene-RETR datasets, our proposed CORE attains **22.99%** Acc\@0.25, achieving more than **4.6× performance improvement**. On the other hand, CORE achieves only **6.24%** Acc\@0.25 on the existing ScanRefer dataset, whereas it reaches **22.99%** Acc\@0.25 on our proposed CrossScene-RETR dataset. These results demonstrate that the proposed method and dataset bring significant performance improvements to the CSSRG task, broadening its practical applicability.
>
> **Q2: Experiments rely on a single 3D backbone; generality to newer encoders is untested.**
>
> **R2:**  We understand your expectation for a comprehensive evaluation. However, we would like to emphasize that our proposed CORE is a general framework for addressing the CSSRG task, into which existing 3D backbones can be seamlessly plugged and utilized. Without loss of generality, we follow the backbone setting adopted in recent 3D Visual Grounding (3DVG) works [A, B, C] to ensure consistency when comparing with 3DVG methods in our comparative experiments. In this setting, we have already adopted newer and task-specific 3D backbones, i.e., Mask3D [D] (2023) for object proposal extraction and Spatial Transformer [E] (2023) for feature encoding. Due to time constraints, we are unable to conduct evaluations on multiple 3D backbones during the rebuttal stage. In the future, we will focus on exploring new 3D backbones to deliver up-to-date applications and generality analysis.
>
> **Q3: The four 3DVG text datasets use heterogeneous text formats, yet the paper gives no preprocessing strategy for syncing them.**
>
> **R3:** Although the text comes from different datasets and has heterogeneous text formats (e.g., .json, .csv, etc.), the processing pipeline described in our paper simply involves using the corresponding text file parsing approaches to load texts from the raw data files and obtaining general embeddings through the pre-trained BERT, following mainstream 3DVG methods [A]. This does not require any specialized text preprocessing or style synchronization, and can directly provide general text embeddings for subsequent use by all methods. In the future, we will provide more details to eliminate any ambiguity, and we will open-source all relevant code after the peer review stage.
>
> **Q4: Scope is limited to indoor static scenes.**
>
> **R4:** Our CORE is a general cross-scene object grounding framework for the CSSRG task, which is not limited to indoor environments and can also be applied to outdoor scenes. Due to the limited availability of suitable outdoor 3D-text multimodal data, most existing 3D object grounding works and our proposed CSSRG are currently focused on these indoor data. Our work is an initial attempt, and in the future, we aim to target outdoor scenarios and build dynamic outdoor CSSRG datasets to support real-world applications such as autonomous driving.
>
> **References:**
>
> [A] Zhu, Ziyu, et al. "3d-vista: Pre-trained transformer for 3d vision and text alignment." Proceedings of the IEEE/CVF International Conference on Computer Vision. 2023.
>
> [B] Jia, Baoxiong, et al. "Sceneverse: Scaling 3d vision-language learning for grounded scene understanding." European Conference on Computer Vision. Cham: Springer Nature Switzerland, 2024.
>
> [C] Huang, Wencan, Daizong Liu, and Wei Hu. "Advancing 3d object grounding beyond a single 3d scene." Proceedings of the 32nd ACM International Conference on Multimedia. 2024.
>
> [D] Schult J, Engelmann F, Hermans A, et al. Mask3D: Mask Transformer for 3D Semantic Instance Segmentation[C]//2023 IEEE International Conference on Robotics and Automation (ICRA). IEEE, 2023: 8216-8223.
>
> [E] Chen, Shizhe, et al. "Language conditioned spatial relation reasoning for 3d object grounding." Advances in neural information processing systems 35 (2023): 20522-20535.

---

> > ### Author Response · Authors · 2025-08-04
> >
> > We would like to know if our response has addressed your concerns. If you have any additional feedback, concerns, or suggestions regarding our manuscript or rebuttal, we would greatly appreciate the opportunity to discuss them further and work on improving the manuscript. Thank you again for the time and effort you dedicated to reviewing this work.

---

> ### Comment · Reviewer_aLPd · 2025-08-04
>
> Dear Authors,
>
> Thank you for taking the time to address my comments. You're response addressed most of my concerns. I have no further questions at this time.

---

> > ### Author Response · Authors · 2025-08-06
> >
> > Dear Reviewer,
> >
> > Thank you for your response. We are pleased to hear that our rebuttal has addressed your concerns. Please don’t hesitate to contact us if you have any further questions.

---

### Note · Authors · 2025-08-13

We sincerely thank the Area Chairs and Reviewers for their time, effort, and thoughtful feedback on our paper. We appreciate the insightful comments and recognition of our work from the reviewers.

Four reviewers acknowledge the strong motivation, novelty, and readability of our paper. More specifically:
- **Reviewers aLPd, Doy1, and Esws** recognize the sound motivation, practicality, and challenging nature of our proposed CSSRG task.
- **Reviewers aLPd and 41F3** affirm the robustness and effectiveness of the proposed cross-modal alignment method, CoRe.
- **Reviewers Doy1 and Esws** appreciate the completeness, significance, and inspiration of our CrossScene-RETR benchmark dataset.

The main concerns focus on the need for more detailed explanations in the Method and Dataset Sections, clearer descriptions and practical relevance of the experimental settings, and further clarification and refinement of several figures.

**During the rebuttal stage, we have provided detailed point-by-point responses to all reviewer comments. All four reviewers have acknowledged that our responses have addressed their concerns.** Specifically, we provide further clarification and analysis of our proposed CrossScene-RETR dataset, CoRe method, and experimental details, supported by our Supplementary Material. Additionally, we carefully proofread the entire manuscript to ensure that all minor issues have been corrected, which will be updated in the future version.

It is an initial attempt to extend the scope of 3D object reasoning to the complete 3D scene set. In the future, we will continue to explore CSSRG in outdoor and dynamic scenarios, embracing the practical significance for real-world applications such as autonomous driving.

---

### Decision · Program_Chairs · 2025-09-17

**Decision:**

Accept (poster)

**Comment:**

The authors propose a new task, Cross-Scene Spatial Reasoning and Grounding (CSSRG), where agents need to find described objects across multiple 3D scenes rather than just predefined ones. To tackle this, they introduce the CoRe framework, which uses robust scene matching and precise object grounding to outperform previous methods while reducing computational costs, as shown through experiments on several standard datasets.

The strengths of this work include a well-motivated and practical introduction to the challenging CSSRG task, the development of a novel cross-modal alignment method, and the introduction of the new CrossScene-RETR benchmark dataset.

During the rebuttal stage, the authors provided detailed responses that effectively addressed the major concerns raised by the reviewers. The reviewers acknowledged the rebuttal, and their final ratings remained consistent. Therefore, the final decision is to accept the paper.